# Bivariate Entropy Analysis of Electrocardiographic RR–QT Time Series

**DOI:** 10.3390/e22121439

**Published:** 2020-12-20

**Authors:** Bo Shi, Mohammod Abdul Motin, Xinpei Wang, Chandan Karmakar, Peng Li

**Affiliations:** 1School of Medical Imaging, Bengbu Medical College, Bengbu 233030, China; shibo@bbmc.edu.cn; 2Department of Electrical and Electronic Engineering, University of Melbourne, Melbourne, VIC 3110, Australia; m.a.motin@ieee.org; 3School of Control Science and Engineering, Shandong University, Jinan 250061, China; wangxinpei@sdu.edu.cn; 4School of Information Technology, Deakin University, Geelong, VIC 3225, Australia; 5Division of Sleep and Circadian Disorders, Brigham and Women’s Hospital, Harvard Medical School, Boston, MA 02115, USA

**Keywords:** cross entropy, joint distribution entropy, RR–QT relationship, ambulatory monitoring

## Abstract

QT interval variability (QTV) and heart rate variability (HRV) are both accepted biomarkers for cardiovascular events. QTV characterizes the variations in ventricular depolarization and repolarization. It is a predominant element of HRV. However, QTV is also believed to accept direct inputs from upstream control system. How QTV varies along with HRV is yet to be elucidated. We studied the dynamic relationship of QTV and HRV during different physiological conditions from resting, to cycling, and to recovering. We applied several entropy-based measures to examine their bivariate relationships, including cross sample entropy (XSampEn), cross fuzzy entropy (XFuzzyEn), cross conditional entropy (XCE), and joint distribution entropy (JDistEn). Results showed no statistically significant differences in XSampEn, XFuzzyEn, and XCE across different physiological states. Interestingly, JDistEn demonstrated significant decreases during cycling as compared with that during the resting state. Besides, JDistEn also showed a progressively recovering trend from cycling to the first 3 min during recovering, and further to the second 3 min during recovering. It appeared to be fully recovered to its level in the resting state during the second 3 min during the recovering phase. The results suggest that there is certain nonlinear temporal relationship between QTV and HRV, and that the JDistEn could help unravel this nuanced property.

## 1. Introduction

Representing ventricular repolarization lability, the beat-to-beat QT interval variability (QTV) in the electrocardiogram (ECG) is an accepted biomarker for the risk of ventricular arrhythmias and cardiac mortality [1,2,3,4]. On one hand, the autonomic nervous system is believed to exert direct influence on the depolarization and repolarization of the ventricle [5,6], making the QTV not simply a substitute of heart rate variability (HRV), a well-established indicator of cardiovascular health [7,8,9,10]. On the other hand, as QT interval readily adapts to the change in heart rate (HR), the QTV is indeed strongly modulated by the HRV. This complicates the interpretation of QT intervals measured at different HRs and consequently, has motivated a line of studies on the correction of QTV for HR [11].

However, the relationship between RR and QT intervals is found to have substantial inter-individual variability [11,12,13], hindering the possibility to have a universally applicable model to formulate the correction of QTV. A more preferable and practical solution is to directly study their relationship. For example, instead of seeking a generalized equation between RR and QT across multiple subjects, prior studies have investigated individualized transfer function to characterize the subject-specific RR–QT dynamical relationship [14,15]. They are so far limited to the use of linear functions such as the auto regressive moving average model. However, as a result of the competition of spontaneity and adaptability [16,17], it is widely accepted that the cardiac activity is nonlinear in nature. The use of nonlinear approach in especially the study of HRV has seen tremendous improvement than traditional linear methods in terms of sensitivity to different physiological or pathological condition [17]. Study of the RR–QT relationship using nonlinear methods remains lacking.

Therefore, we sought to examine the RR–QT nonlinear relationship in this current pilot study. Motivated by recent progressions in entropy measures in analyzing limited-length physiological signal [16,17,18,19,20,21], we were mainly focusing on bivariate entropy analysis. To study how the RR–QT nonlinear relationship adapts to different physiological states, we analyzed ECG data collected before in a small number of physically- and mentally healthy college students under a resting-cycling-recovering protocol. Detailed algorithms of the bivariate entropy analysis were specified in Section 2. New findings from this work were summarized in Section 3 and were followed by discussions in the last section.

## 2. Materials and Methods

### 2.1. Data and Subjects

We analyzed ECG data of 19 college students (10 females/9 males) aged 20.9 ± 1.1 years [mean ± standard deviation (SD)] collected in a previous study approved by the Ethics Committee in Clinical Study of Bengbu Medical College with written informed consent obtained from all subjects. They were free from cardiovascular conditions including histories and were mentally healthy, confirmed by questionnaires. For each subject, ECG was recorded using a Holter monitor (HeaLink-R211B, HeaLink Ltd., Bengbu, China) under three different conditions—resting while seated on a chair (2 min), cycling on a treadmill (3 min), and seated recovering (10 min). The sampling frequency was 400 Hz and the standard unipolar chest lead V5 was applied.

### 2.2. Data Preprocessing

Raw ECG recordings under each condition were first filtered with a second-order infinite impulse response notch filter for removing power inference (i.e., 50 Hz). Baseline wander of ≤1 Hz was then removed based on wavelet decomposition.

R-peaks from each ECG recording were identified using a template matching approach [22] and after that, a semi-automated template compressing-stretching algorithm [23] was used to extract QT intervals. This algorithm requires the operator to define a template QT interval by selecting the beginning of the Q wave and the end of the T wave for one beat of the ECG signal. The algorithm then finds the QT interval of all other beats by calculating how much each beat must be temporally stretched or compressed to best match the template interval. Figure 1 illustrates the identified R-peaks and the extracted RR and QT intervals under all three conditions for a typical subject.

A non-stationary trend of ≤ 0.04 Hz was removed from each RR and QT time series using a wavelet decomposition-based algorithm [24]. Anomalous intervals manifested as spikes in the time series due to either false negatives or false positives during the R-peak detection or caused by erratic compressing/stretching during the QT intervals extraction were sporadically seen. To diminish the effect of these spikes on entropy analysis, we applied a two-step procedure to minimize them (see Figure 2). In the first step, an impulse rejection filtering was performed with spikes identified being replaced by the median value of surrounding five samples [25] (Figure 2 left panel). In the second step, a moving standard deviation with window length of 100 samples was performed and any points that were three standard deviations away from the global mean were removed (Figure 2 middle panel). We should note that in the second step, points identified as spikes in one of two time series (i.e., RR or QT intervals) were removed simultaneously from the other time series. The right panel in Figure 2 shows an example of the final RR and QT intervals time series to be analyzed.

### 2.3. Bivariate Entropy Analysis of RR–QT Intervals Time Series

To assess the QT–RR relationship, we analyzed the bivariate entropy that quantifies essentially the beat-to-beat cross predictability or synchrony between QT and RR interval time-series. Four different bivariate entropy algorithms as specified below were used.

#### 2.3.1. Cross Sample Entropy (XSampEn)

For a pair of normalized time-series (i.e., zscored) u(i) and v(i), i=1,2,⋯,N, the corresponding state space representations are:$$\left\{\begin{array}{c} X_{m}\left(i\right)=\left[u\left(i\right),u\left(i+\tau \right),\cdots ,u\left(i+\left(m-1\right)\tau \right)\right]\\ Y_{m}\left(j\right)=\left[v\left(j\right),v\left(j+\tau \right),\cdots ,v\left(j+\left(m-1\right)\tau \right)\right] \end{array}\right.,$$
where 1≤i,j≤N−mτ, with *m* being the embedding dimension and τ the time delay parameter. Define the distance between the the two vectors $X_{m}\left(i\right)$ and $Y_{m}\left(j\right)$ by $d_{i,j}^{\left(m\right)}=\left|X_{m}\left(i\right),Y_{m}\left(j\right)\right|$, where |·| indicates the maximum norm. Denote $A_{i}^{\left(m\right)}\left(r\right)$ the average number of *j*’s that meet $d_{i,j}^{\left(m\right)}\leq r$ for all j=1,2,⋯,N−mτ, where *r* is a predefined threshold parameter. In similar means $A_{i}^{\left(m+1\right)}\left(r\right)$ can be defined as *m* incremented by 1. The XSampEn of the two time-series can be calculated by [19]:(1)$$\mathrm{XSampEn}\left(m,\tau ,r\right)=-\ln\frac{\sum_{i=1}^{N-m\tau }A_{i}^{\left(m+1\right)}\left(r\right)}{\sum_{i=1}^{N-m\tau }A_{i}^{\left(m\right)}\left(r\right)}.$$

#### 2.3.2. Cross Fuzzy Entropy (XFuzzyEn)

XFuzzyEn is actually a refined algorithm of XSampEn. The difference between them lies in the thresholding procedure. Specifically, instead of calculating the average number of *j*’s that meet $d_{i,j}^{\left(m\right)}\leq r$ as shown in Section 2.3.1, XFuzzyEn calculates the average degree of membership, i.e., the $A_{i}^{\left(m\right)}\left(r\right)$ is defined by [20]:(2)$$A_{i}^{\left(m\right)}\left(r\right)=\frac{1}{N-m\tau }\sum_{j=1,j\neq i}^{N-m\tau }e^{-\ln\left(2\right){\left(\frac{d_{i,j}}{r}\right)}^{2}},$$
and so is $A_{i}^{\left(m+1\right)}\left(r\right)$. XFuzzyEn can then be defined by replacing $A_{i}^{\left(m\right)}\left(r\right)$ and $A_{i}^{\left(m+1\right)}\left(r\right)$ in (Equation 1) with the new definitions here. It has been shown in published studies that the introduction of fuzzy membership function significantly improves the stability and consistency of the algorithm [20,26,27,28].

#### 2.3.3. Cross Conditional Entropy (XCE)

XCE estimates how predictable it is for a corresponding new sampling point in a target sequence given *m* samples in a template sequence, in other words how much new information carried by this new sampling point. To be specific, the full range of dynamics of both the target and template sequences, i.e., v(j) and u(i), is first divided into a fixed number of ξ values labelled from 0 to ξ-1. The variable ξ is named quantification level and here [max(u,v)−min(u,v)]/ξ represents the coarse-graining resolution. This coarse-graining process renders u(i) and v(j) sequences of symbols which are denoted here by $\hat{u}\left(i\right)$ and $\hat{v}\left(j\right)$. The two sequences of symbols are then reconstructed to
$$\left\{\begin{array}{c} X_{m}\left(i\right)=\left[\hat{u}\left(i\right),\hat{u}\left(i-\tau \right),\cdots ,\hat{u}\left(i-\left(m-1\right)\tau \right)\right]\\ {\mathrm{YX}}_{m+1}\left(j\right)=\left[\hat{v}\left(j\right),X_{m}\left(j\right)\right] \end{array}\right.,$$
where (m−1)τ+1≤i,j≤N. Codify the vectors X and YX in decimal format as
$$\left\{\begin{array}{c} w_{i}=\hat{u}\left(i\right)\xi^{m-1}+\hat{u}\left(i-\tau \right)\xi^{m-2}+\cdots +\hat{u}\left(i-\left(m-1\right)\tau \right)\xi^{0}\\ z_{j}=\hat{v}\left(j\right)\xi^{m}+w_{i} \end{array}\right.,$$
which renders the two vector sequences X and YX series of integer numbers $w_{i}$ and $z_{j}$ with $w_{i}$ ranging from 0 to $\left(\xi -1\right)\sum_{i=1}^{m-1}\xi^{i}$, and $z_{j}$ from 0 to $\left(\xi -1\right)\sum_{j=1}^{m}\xi^{j}$. The frequency of each possible value in the two integer sequences can then be easily obtained. The XCE is then calculated by [18]:(3)$$\mathrm{XCE}\left(m,\tau ,\xi \right)=\mathrm{SE}\left(z_{j}\right)-\mathrm{SE}\left(w_{i}\right)+\mathrm{perc}\left(m\right){\mathrm{SE}}_{v}\left(1\right),$$
where SE(·) represents the Shannon entropy; perc(m) is the percentage of patterns $w_{i}$ that are found only once; ${\mathrm{SE}}_{v}\left(1\right)$ means the Shannon entropy of the quantized sequence $\hat{v}\left(j\right)$.

#### 2.3.4. Joint Distribution Entropy (JDistEn)

In JDistEn, the distribution pattern of $d_{i,j}^{\left(m\right)},1\leq i,j\leq N-m\tau ,i\neq j$ is considered. Specifically, the empirical probability density function of $d_{i,j}^{\left(m\right)}$ is estimated by histogram with a predefined bin number *B*, which is denoted by $p_{t}$ where t=1,2,⋯,B. Then JDistEn is defined by the Shannon formula for entropy [21]:(4)$$\mathrm{JDistEn}\left(m,\tau ,B\right)=-\frac{1}{\log_{2}\left(B\right)}\sum_{t=1}^{B}p_{t}\log_{2}\left(p_{t}\right).$$JDistEn has been shown to have good performance in especially short-length data [21].

#### 2.3.5. Parameter Selection

We applied the commonly accepted parameters for computing these entropy measures. Specifically, for embedding dimension, m=2 was used; τ=1 was used for the time delay parameter; r=0.2 was used for the threshold value for calculating XSampEn and XFuzzyEn; ξ=6 was used for XCE calculation; and B=512 was used for the bin number parameter in calculating JDistEn.

In addition, to eliminate the potential effect of data length on entropy results, we split the data during recovering stage into three sections: recovering phase 1 defined as the first three min, recovering phase 2 defined as the period between (3, 6] min during the recovering, and recovering phase 3 defined as the period between (6, 9] min during the recovering.

### 2.4. Statistical Analysis

Shapiro–Wilk *W*-tests accepted the normal distribution hypothesis for all four studied bivariate entropy measures. Linear mixed effects models examined the difference in each of the studied measures across resting, cycling, and the three recovering stages. Age and sex were adjusted. These statistical analyses were performed using MATLAB (Ver. R2020a, the MathWorks Inc., Natick, MA, USA).

## 3. Results

Figure 3 and Table 1 summarize the main results of this study. No statistically significant changes were found in XSampEn, XFuzzyEn, or XCE during different conditions, except that the XCE during recovering phase 2 was significantly greater than that during resting state [p=0.0004; Figure 3(C1); Table 1]. JDistEn demonstrated a decrease during cycling compared with the resting state (p=0.001; Table 1), and then a recovering increase from cycling to recovering phase 1 (p=0.09 in post hoc comparison between recovering phase 1 and cycling), and further to phase 2 (p<0.0001 in post hoc comparison between recovering phase 2 and cycling). During recovering phase 2, JDistEn was fully recovered to its level in resting state [p=0.7 in post hoc comparison between resting and recovering phase 2; Figure 3(D1)].

The changes in JDistEn also appeared to be quite consistent across different subjects as shown in Figure 3(D1). Specifically, of the 19 subjects, 15 demonstrated decreases whereas four subjects showed increases in JDistEn during cycling. Again, 15 of them showed consistent recovering of JDistEn levels from recovering phase 1 to recovering phase 2 whereas only four subjects showed opposite changes.

## 4. Discussion

In the current study, we examined the within-subject changes in beat-to-beat electrocardiographic QT–RR relationship in response to a resting-cycling-recovering test protocol. Different from prior studies that had focused on formulating the relationship with an explicit equation, such as a polynomial function, we studied the cross-predictability or synchrony between QT and RR interval time series assessed by four bivariate entropy measures that characterized how the two processes vary along with each other. While three of them did not demonstrate observable changes across the three conditions, the JDistEn showed a consistent decline during cycling compared with resting state; it also demonstrated a recovering trend during the three recovering phases.

As this current pilot study is exploratory, we did not have a prior hypothesis regarding how this bivariate relationship changed under different conditions. There are thus two scenarios in interpreting our findings. On the first, the apparent difference between JDistEn and the other measures may be indicative of the superiority of JDistEn over the remained three in terms of capturing weak changes in the synchrony between two short recordings. This advantage has been well demonstrated in a previous paper using experimental data and in a theoretical perspective [21]. The ability of capturing weak changes also implies a better sensitivity of JDistEn that has been proved as well in the previous study using simulated bivariate processes with known coupling behavior [21]. On the second, our results based on JDistEn indeed suggest a changed QT–RR dynamical relationship in response to physiological stresses, reiterating or backing up the knowledge that the change in cardiac depolarization-repolarization does not simply mimic or reflect the change in heart rate, but receives director modulating from upstream controls such as the autonomic balance.

Physiological interpretation of QT interval changes in response to changes in heart rates under stresses such as during walking is not fully understood yet. It has been reported that rapid heart rate change could lead to transient dissociation of ventricular paced QT variation from RR variation [29]. Besides, there exists a hysteresis in QT–RR adaptation [29]. It may take several minutes for QT intervals to achieve a steady state after the change of heart rate. In addition, other physiological factors such as autonomic tone also affect QT interval independently from chronotropic modulation of the sino-atrial node. Therefore, our observed QT–RR relationship during cycling might be a joint effect from both heart rate modulation on ventricular repolarization and the direct autonomic influence of the ventricular myocardium [30], or, altogether, the differential onsets of autonomic regulation of the sino-atrial node and ventricular myocardium [31].

Bivariate entropy method is used to measure the degree of synchrony between two time series [19,21]. If a pair of series is more synchronous than another pair, it should have lower values of cross entropy and higher value of JDistEn [19,21]. Cross entropy measures other than JDistEn has shown an statistically insignificant increase in values during the cycling phase of the experiment and decreasing during first recovering phase. However, the subsequent recovering phases showed random behaviour (increasing-decreasing) rather than any specific trend. In contrast, results of JDistEn showed a statistically significant drop during cycling, indicating that the RR and QT series becomes less synchronized during cycling than in the rest. The value of JDistEn increases in subsequent two recovery phases, which again indicates that the perturbed synchronization that is caused during cycling is recovering with the progressing of resting phase. Although there is a little drop in JDistEn value at the last part of recovery phase, that can be described as a damping effect before stabilizing the bivariate entropy values in resting phase. However, analysis on longer signal is necessary to support these findings.

We did not correct QT intervals based on heart rate variations in the current work. The existing approaches for QT correction including for example the Bazett or Fridericia approaches are almost all based on parabolic regression models (i.e., QT = β× RR${}^{\alpha }$). There are studies indicating that the optimal coefficient (i.e., α) could vary significantly from 0.23 to 0.49 [11], implying that a correction based on either of these universal equations might be problematic [11,13]. In the context of the relationship analysis between QT and RR, the interpretation of results may become more challenging after correcting the QT intervals.

We should also note that our results were based on a rather small size of samples. Further studies with larger samples are warranted to examine the generalization capability of our results. A previous study has showed that, in general, women had steeper and more curved QT–RR patterns than men [12], implying that a sex difference might be existing and worthy of further explorations. Besides, we were not trying to examine the value of these bivariate entropy metrics for differentiating different physiological states in this exploratory pilot study, further studies in a more diverse population (e.g., with aged or diseased participants) are warranted in order to understand whether these differences still exist or change (i.e., either diminish or exacerbate), such that they could possibly be used a biomarker for biological aging and/or certain pathological processes. In addition, our analysis was based on single lead ECG (i.e., V5). It has been reported that QTV differs across leads [32]. Further studies may take full advantage of multi-lead wearable devices and examine and confirm the RR–QT changes in daily living conditions across different leads.

## Figures and Tables

**Figure 1 entropy-22-01439-f001:**
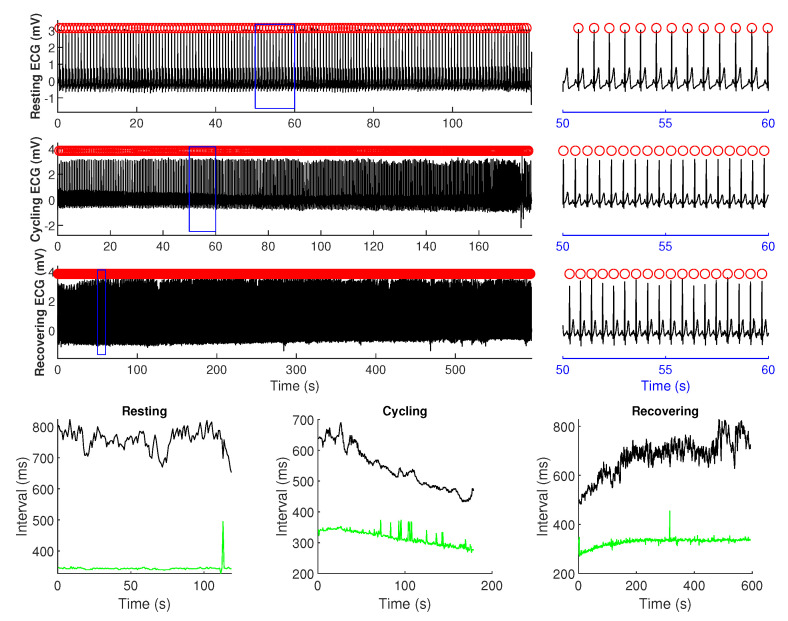
RR and QT intervals from a representative subject under resting, cycling, and recovering conditions. Top three panels show the electrocardiogram (ECG) data during resting, cycling, and recovering conditions with R-peaks marked in *red circle*. A zoomed portion (marked by *blue box*) is also shown next to each panel on the right-hand side for better visualization. Bottom panels visualize the RR and QT interval time series under each condition.

**Figure 2 entropy-22-01439-f002:**
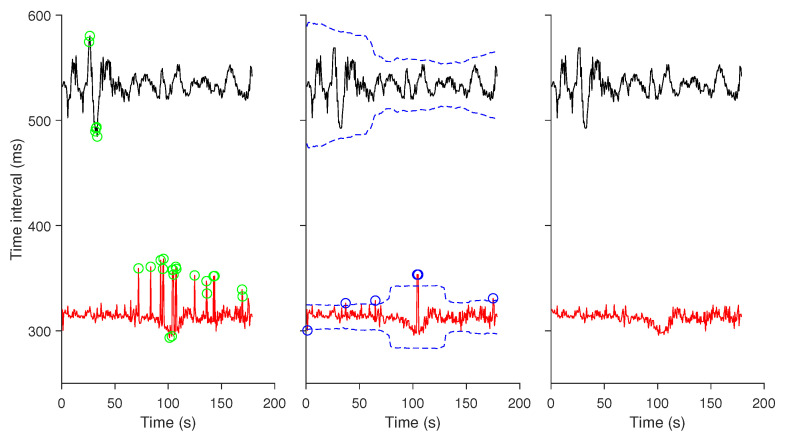
Parallel removal of anomalous intervals from RR and QT intervals time series. The time series were from the same subject demonstrated in Figure 1 during cycling. Trends were removed from each of them. An impulse rejection filtering was applied to identify spikes (*green circle*) that were subsequently replaced with the median value of surrounding five samples (*left panel*). A moving standard deviation with window size of 100 was performed and for each point two threshold values (*blue dashed line*) that were above or below three times the corresponding standard deviation from the global mean were used to screen extremes (*blue circle*) for the second time (*middle panel*). The points corresponding to those extremes were removed simultaneously from both RR and QT intervals time series (*right panel*).

**Figure 3 entropy-22-01439-f003:**
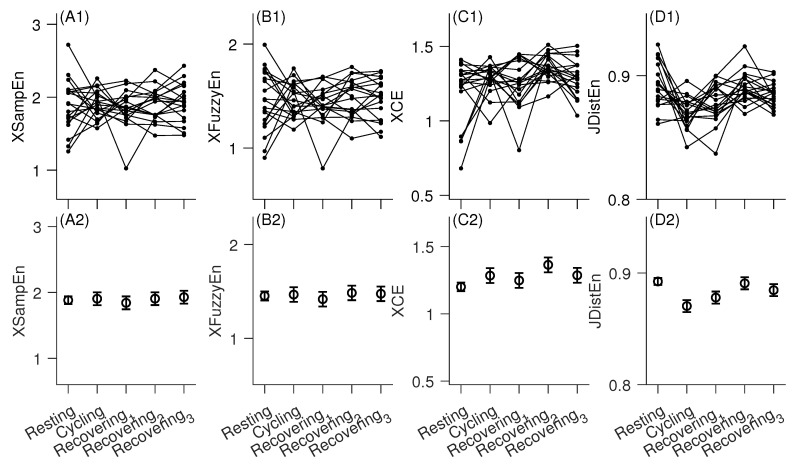
Bivariate entropy analysis of RR–QT intervals time series. (**A1**–**D1**) Raw data are shown by dots and results from the same subject are connected using *line*. (**A2**–**D2**) Model estimates for each measure under different conditions are shown as mean (*circle*) and standard error (*error bar*).

**Table 1 entropy-22-01439-t001:** Bivariate entropy analysis of RR–QT time series. Shown results from linear mixed effects models with adjustment of age and sex.

	XSampEn		XFuzzyEn		XCE		JDistEn	
Variable	Mean ± SE	*p*	Mean ± SE	*p*	Mean± SE	*p*	Mean ± SE	*p*
Intercept	1.91 ± 0.07	-	1.47 ± 0.05	-	1.19 ± 0.04	-	0.892 ± 0.003	-
Sex (male)	−0.05 ± 0.07	0.51	−0.04 ± 0.06	0.49	0.03 ± 0.03	0.44	0.002 ± 0.003	0.60
Age	−0.02 ± 0.03	0.62	−0.01 ± 0.03	0.61	0.00 ± 0.01	0.83	−0.003 ± 0.001	0.01
Cycling	0.02 ± 0.08	0.78	0.01 ± 0.06	0.82	0.08 ± 0.04	0.07	−0.022 ± 0.004	<0.0001
Recovering phase 1	−0.04 ± 0.08	0.60	−0.03 ± 0.06	0.58	0.05 ± 0.04	0.28	−0.014 ±0.004	0.001
Recovering phase 2	0.02 ± 0.08	0.77	0.03 ± 0.06	0.60	0.16 ± 0.04	0.0004	−0.002 ± 0.04	0.70
Recovering phase 3	0.05 ± 0.08	0.55	0.02 ± 0.06	0.71	0.08 ± 0.04	0.06	−0.008 ± 0.004	0.07

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
