# Peer review of "Bivariate Entropy Analysis of Electrocardiographic RR–QT Time Series"

_entropy, 2020, doi:10.3390/e22121439_

Round 1

Reviewer 1 Report

The article describes application of entropy measures to signal analysis.
More precisely, the Authors applied four bivariate entropy measures (cross sample entropy, cross fuzzy entropy, cross conditional entropy, and joint distribution entropy) to assess the QT-RR relationship in ECG signals. Different physiological conditions were considered in the research: resting, cycling and recovering. In case of the first three entropy measures, no statistically significant differences across these conditions were observed. The differences were observed for the joint distribution entropy.

The paper represents good quality, detailed comments are given below:
1) In Figure 1, it would be good to describe the units of the vertical axes.
2) In the description of the Figure 2, probably better is to replace "extremities" with "extremes".
3) In line 109, maybe better is to replace "Fig. 3" with "Fig. 3 (C1)”. In lines 110-115, maybe better is to refer to "Fig. 3 (D1)" and only once.
4) In Table 1, instead of "CCE" should be "XCE".
5) Should the values in Table 1 correspond to those in Figure 3?
6) In the References section, the titles of journals sometimes use capital letters and sometimes lowercase letters.

The above minor comments do not affect the generally high quality of the paper. In my opinion, after these minor corrections, the article should be published in the Entropy Journal. Especially that its topic perfectly matches the title of the journal.

Reviewer 2 Report

This is interesting work that suggests the authors' novel application of bivariate entropy analysis might have arrhythmia predictive.  The authors should emphasize the fact that this study was purely exploratory and that it provides no proof of the sought after predictive value.  Ultimately, this must be tested in an at-risk population.
